# YaoGAN: Learning Worst-case Competitive Algorithms from Self-generated Inputs

## Abstract

We tackle the challenge of using machine learning to find algorithms with strong worst-case guarantees for online combinatorial optimization problems. Whereas the previous approach along this direction (Kong et al., 2018) relies on significant domain expertise to provide hard distributions over input instances at training, we ask whether this can be accomplished from first principles, i.e., without any human-provided data beyond specifying the objective of the optimization problem. To answer this question, we draw insights from classic results in game theory, analysis of algorithms, and online learning to introduce a novel framework. At the high level, similar to a generative adversarial network (GAN), our framework has two components whose respective goals are to learn the optimal algorithm as well as a set of input instances that captures the essential difficulty of the given optimization problem. The two components are trained against each other and evolved simultaneously. We test our ideas on the ski rental problem and the fractional AdWords problem. For these well-studied problems, our preliminary results demonstrate that the framework is capable of finding algorithms as well as difficult input instances that are consistent with known optimal results. We believe our new framework points to a promising direction which can facilitate the research of algorithm design by leveraging ML to improve the state of the art both in theory and in practice.

## 1 Introduction

It has been a long-standing ambition of machine learning to learn from *first principles*, that is, a machine learning program that learns *without any human-provided data* beyond an appropriate definition of the task at hand (Silver et al., 2018). The community has made impressive strides towards this goal in recent years, most notably in the domain of tabletop games such as chess, Go, shogi (Silver et al., 2018) and poker (Brown & Sandholm, 2018), where programs have been trained to play at superhuman levels without consuming any game-specific human expertise except, naturally, the rules of the game.

In this work, we look at the task of using ML to design algorithms for optimization problems. Similar to tabletop games, the task of algorithm design has succinct and well-defined objectives, so learning from first principles is potentially feasible. Moreover, the task is very intriguing as it typically requires significant amount of time and expertise for humans to accomplish.

In our context, the problem of learning algorithms from first principles is also related to an important issue in both ML and algorithms: what input instances is the learned algorithm expected to handle? The typical ML approach is to train the program against a pre-defined training set, with the assumption that the training set is carefully chosen to capture future input distributions well, so the learned algorithm generalizes to future input of interest. On the other hand, the algorithm design community in theoretical computer science (TCS) traditionally considers algorithms with worst-case guarantees due to their robustness against arbitrary input. We focus on the latter domain in our work.

Along the line of learning worst-case algorithms, a particularly relevant result is by Kong et al. (2018) that uses reinforcement learning to learn theoretically optimal algorithms for online optimization problems, and one of the key ideas they introduce is the notion of universal and high-entropy training sets. Roughly speaking, such a training set serves as a very effective benchmark in the sense that the task of finding an algorithm with strong worst-case guarantee reduces to finding

an algorithm that performs well against this training set. Not surprisingly, for a given optimization problem, coming up with such a training set can require significant domain expertise a priori. In the current work the goal is to go beyond the framework of universal training sets, and enable learning of algorithms without even this prior domain knowledge.

To overcome this issue, we draw insights from the fields of game theory, analysis of algorithms, and online learning, as well as generative adversarial networks (GANs, Goodfellow et al. (2014)). In particular, we introduce a training framework that aims to learn a strong worst-case algorithm and an effective benchmark training set simultaneously, all without relying on any human domain expertise beyond specifying how to compute the objective of the optimization problem. If successful, such an approach would have impactful ramifications: researchers could accelerate the speed of discovery by offloading to machines the time-consuming process of finding problem-specific insights, which can be crucial in improving the state of the art for many problems both in theory and in practice.

At the high level, our framework has two components: the algorithm agent and the adversary. Their goals are to learn the optimal algorithm and a set of input instances that captures the essential difficulty of the given optimization problem, respectively. For simple optimization problems, rudimentary strategies such as random search or grid search can provide a fairly effective adversary. For problems with more interesting underlying space, we exploit deep learning to guide our search for the distribution of difficult inputs. Similar to a generative adversarial network (GAN), we use two neural networks, one for the algorithm agent and another for the adversary. We start from scratch, train these two networks against each other, and let them evolve simultaneously. We call our program *YaoGAN* due to this structural similarity to GAN and the connection to the classic Yao's Lemma (see Section 2.1 for details); this is our main technical contribution.

In this initial effort, we follow Kong et al. (2018) in focusing on online optimization problems (Buchbinder et al., 2009) where each input is a sequence arriving over time, and the algorithm needs to make an immediate and irrevocable decision at each time step without knowing the future input. We further restrict our attention to problems that can be solved or approximated by algorithms requiring a small amount of memory. There are abundant well-studied and interesting problems bearing such properties in online optimization, typically when the structure of the problem exhibits certain Markov property so it is sufficient for the algorithm to maintain a succinct state capturing past information.

We consider two online optimization problems, the *ski rental* problem and the *fractional AdWords* problem. Ski rental (Karlin et al., 1986) has been a staple introductory example in online analysis. The second problem, fractional AdWords (Mehta et al., 2007), is (a fractional relaxation of) a well-known problem motivated by online advertising. Moreover, the Adwords problem is closely related to numerous variants of the online bipartite matching problem, which is still an actively studied area in algorithm design, for example in the setting of ride-sharing platforms (Dickerson et al., 2017; Ashlagi et al., 2019). For these problems, we have a good understanding of the structures of the optimal algorithm and difficult inputs, so we can verify that our paradigm leads to something meaningful. Our experiments, although fairly limited and preliminary, demonstrate that our framework can be effective at finding algorithms as well as difficult input instances that are consistent with optimal results. We believe further study along this direction can lead to more exciting developments.

We use generic neural network architecture and ML methods (e.g., Adam optimizer (Kingma & Ba, 2014)) in our experiments, and do not optimize the training set-up or hyperparameters, since our focus in this paper is not in extending ML techniques, but in applying our high-level paradigm to learn algorithms from first principles. We believe the co-evolution of new ML techniques tailored to this task will be essential to the ultimate success of this approach. We leave that aspect, as well as applying the framework to more open algorithm design problems, to future work.

**Related work.** The most relevant paper to ours is an approach by Kong et al. (2018) that uses reinforcement learning (RL) to find competitive algorithms for online optimization problems including the AdWords problem, online knapsack and the secretary problem. They introduce the notions of universal and high-entropy training set that are tailor-made to facilitate the convergence to worst-case competitive algorithm. Deudon et al. (2018) and Kool et al. (2018) use RL to solve the traveling salesman problem (TSP) in the two-dimensional plane. Both uniformly sample the input points from the unit square and teach the network to optimize the expected performance on this input distribution without having access to exact solutions. Nevertheless, the learned algorithms from both approaches

seem competitive with state-of-the-art methods, at least for modest input sizes. Vinyals et al. (2015) introduce the Pointer Network, a sequence-to-sequence model that successfully manages to approximately find convex hulls, Delaunay triangulations and near-optimal planar TSPs. They assume access to an exact solution and use supervised learning to train the model, while their input is uniformly sampled from the unit square. Graves et al. (2014) introduce the Neural Turing machines, a differentiable and computationally universal Turing machine model, and use supervised learning to teach them simple algorithms such as copying or sorting. Kaiser & Sutskever (2015) introduce the parallel and computationally universal Neural GPU and use supervised learning to teach it addition and multiplication that seemingly generalizes to arbitrary lengths.

We point out some other related results without going into details: the work of Bello et al. (2016) studies applying RL on combinatorial problems including TSP and Knapsack; the work of Dai et al. (2017) exploits embedding and RL to design heuristic algorithms for classic graph problems on specific distributions; Boutilier & Lu (2016) uses RL to solve Budget Allocation problem; Duetting et al. (2019) applies deep learning techniques to designing incentive compatible auction mechanisms. Readers insterested in ski rental and AdWords can consult Buchbinder & Naor (2009) and Mehta et al. (2013) for a more thorough coverage.

To the best of our knowledge, previous results mostly focus on the algorithm side, and either learn algorithms designed to work on specific input distributions of interest, or utilize supervised learning to mimic a known method, or (in the case of Kong et al. (2018)) rely on domain expertise to provide hard input instances to train algorithms that are competitive in worst-case. We do not wish to detract from the value of optimizing an algorithm on a specific distribution if it is of special interest. Indeed, a practitioner optimizing for a distribution $\mathcal{D}$ could likely design algorithms that outperform classic solutions on $\mathcal{D}$ since classic solutions typically protect against worst-case inputs and might be overly pessimistic on $\mathcal{D}$. Our goal is different: facilitate the discovery of new algorithms with strong worst-case performance on open problems where we do not have a known difficult input distribution at hand. Such learned strategies could then potentially be later fine-tuned to adapt to specific input distributions, while retaining some worst case guarantees.

## 1.1 SUMMARY OF RESULTS

As a brief introduction, we can summarize YaoGANs as essentially starting with a "random" algorithm network, then alternating between (1) finding the *best response* (that will force the algorithm to exhibit its worst behavior), and (2) training the algorithm network on this best response. Details are given in Section 3.

We apply the YaoGAN method to two classic problems in the area of online optimization: the *ski rental* problem and a fractional version of the *AdWords* problem (these are defined formally in Section 2). The two problems exhibit a progression in terms of how hard it is to find the "best responses." Finding best response inputs for ski rental is relatively easy—an approximately exhaustive search (over a suitably parameterized space) is sufficient—while for AdWords it seems to be a hard non-convex global optimization problem.

**Ski rental.** A few minutes of training leads to an algorithm whose *competitive ratio*—the primary figure of quality for an online optimization algorithm—converges approximately to 1.59, which is within $10^{-2}$ of the theoretically best value achievable. Furthermore, the behavior of trained algorithm exactly matches the one of Karlin et al. (1986) (Appendix C).

**Fractional AdWords.** For the fractional AdWords problem—a "budgeted bipartite matching problem"—with 3 advertisers with a budget of 3 each and 9 ad slots[1], about 10 hours of training is sufficient to achieve a competitive ratio of 0.73, which is within $10^{-2}$ of the theoretically best value achievable for this problem. The trained algorithm qualitatively recovers both the greedy strategy and the balance strategy (Kalyanasundaram & Pruhs, 2000), two defining aspects of the optimal solution, as well as seems to find the correct form of tradeoff between the strategies in tested scenarios (details in Section 4).

---

[1] Note that while the problem solved here is defined on "fixed size" instances (27 real numbers), this is already well beyond the regime of executing a simple exhaustive search to discover the optimal algorithm.

## 2 PRELIMINARIES

We start with a discussion of online algorithms and the classic theoretical results upon which we draw insights for our approach. We will keep the discussion high-level and informal since we use these results mostly for intuition rather than extending them to prove any theoretical guarantee.

We chose online optimization problems as the test bed for our approach because they are rich in structure and application. Online problems comprise a theoretical framework to study tasks in the interactive setting. In particular, an algorithm agent is given a sequence of requests that require a response without knowing future requests. A successful algorithm typically needs to hedge between receiving an immediate reward and allowing for potentially lucrative future rewards. The *competitive ratio* of an online algorithm for a particular input instance is defined as the ratio of the (expected) objective value the online algorithm gets for the input over the optimal value an algorithm can receive for the instance if it knows the entire input upfront (i.e., the offline optimal). The worst-case competitive ratio of an online algorithm is its competitive ratio for the worst possible input. We usually drop the "worst-case" when we talk about the competitive ratio of an algorithm. For audience not familiar with the field of online algorithm, we include an expanded background discussion in Appendix A.

Deep learning approaches typically train a tunable network on a particular distribution of interest. When applied to learning algorithm with worst-case guarantees, it is not clear apriori what distribution should one train on. Instead, we take a game-theoretic approach that formalizes the interplay between the algorithm and its inputs.

### 2.1 GAME-THEORETIC VIEW

Consider a (possibly infinite) two-player zero-sum game matrix $V$, where we denote the row player as the *algorithm player*, and the column player as the *adversary* who determines the inputs to the algorithm. Each row represents a deterministic algorithm $\mathcal{A}$, and each column represents a specific input instance $\mathcal{I}$, and the corresponding entry $V(\mathcal{A}, \mathcal{I})$ specifies the payoff to the adversary if the algorithm player plays strategy $\mathcal{A}$, and the adversary plays strategy $\mathcal{I}$. We follow the custom that the algorithm player aims to minimize the payoff to the adversary, while the adversary tries to achieve the opposite. In the case of online algorithms, if we are working with a problem with minimization objective, the payoff $V(\mathcal{A}, \mathcal{I})$ will be the competitive ratio of $\mathcal{A}$ for input $\mathcal{I}$. Similarly, for a maximization problem, we take $V(\mathcal{A}, \mathcal{I})$ as the negation of the above competitive ratio.

**Yao's Lemma.** A randomized algorithm can be viewed as a distribution $\boldsymbol{p}$ over rows of $V$ (i.e., deterministic algorithms); a input distribution $\boldsymbol{q}$ can be viewed as a distribution over columns of $V$ (i.e., pure inputs). The optimal randomized algorithm $\boldsymbol{p}^*$ is one whose expected payoff against the worst-case input is minimized, i.e., $\boldsymbol{p}^* = \arg\min_{\boldsymbol{p}} \max_{\mathcal{I}} \boldsymbol{p}^T V(:, \mathcal{I})$. A standard result in game-theory stipulates that in this case maximizing over pure inputs is equivalent to maximizing over input distributions. The namesake of our paper, Yao's Lemma (Yao, 1977) (see also Borodin & El-Yaniv (1998)) says that there is an adversarial distribution $\boldsymbol{q}^*$ over inputs such that the expected payoff of the best deterministic algorithm over $\boldsymbol{q}^*$ is the same as the value of the best randomized algorithm $\boldsymbol{p}^*$ over its worst possible input. In other words, $\max_{\mathcal{I}}(\boldsymbol{p}^*)^T V(:, \mathcal{I}) = \min_{\mathcal{A}} V(\mathcal{A}, :)\boldsymbol{q}^*$. Using classic game theory notations, we also refer to the best randomized algorithm as the row player's minmax strategy, and $\boldsymbol{q}^*$ the column player's maxmin strategy.

**Solving an online optimization problem** corresponds to solving a zero-sum game, i.e., finding both $\boldsymbol{p}^*$ and $\boldsymbol{q}^*$. Prior work (Kong et al., 2018) explores learning online algorithms by training only the $\boldsymbol{p}^*$ side. They address the $\boldsymbol{q}^*$ by considering only problems for which the optimal input distribution is known. However, there are two acknowledged weaknesses in this approach:

First, such knowledge of the adversarial maxmin distribution is typically only available for well-studied problems where we already have a theoretically established limit on the performance of any algorithm. Thus, to fulfill the ultimate goal of using machine learning to discover (unknown) algorithms with competitive "worst-case" performances, we need an approach without relying on domain knowledge of a worst-case input distribution.

Second, a somewhat fundamental issue with the approach is that an algorithm that performs well against the adversarial distribution alone is in fact not necessarily the optimal algorithm. For a

simple example, consider the game of rock-paper-scissors. The maxmin column strategy is to play the three moves uniformly at random, and any row strategy (e.g., always play rock) is a best response to the column player's minmax strategy. Indeed, this was acknowledged in Kong et al. (2018), e.g., for the AdWords problem they combined the adversarial distribution with another hand-picked input distribution to avoid converging to a suboptimal greedy algorithm.

As mentioned earlier, our approach avoids these two drawbacks by not trying to come up a good training set upfront, and instead aims to solve for the algorithm's maxmin solution directly using the framework of no-regret dynamics B, by co-training the adversary and the algorithm.

## 2.2 SKI RENTAL

We formally define the ski rental problem and illustrate the notions of deterministic strategies and competitive ratios. Randomized and optimal strategies are described in Appendix C.

**Problem 1 (Ski rental)** *Suppose we want to go skiing in the winter, which requires a set of skis. Each day we go skiing, we have two options: either rent the skis for $\$1$, or buy the skis for $\$B$, for some fixed $B$. Naturally, renting the skis allows us to ski only for that one day, while after buying we can use them for an unlimited number of days. The crux is that we do not know in advance the total number of days $k$ we will be able to ski (e.g., the weather might change, we might break a leg, etc.). The objective is to minimize the total amount of money spent by strategically buying/renting skis.*

**CR of deterministic strategies.** A deterministic strategy $\mathcal{A}$ takes the form of renting for up to some fixed number $x$ of days initially, and if it turns out that we go skiing for at least $x$ days, we buy on day $x$. If the eventual number of days we go skiing is $k$, the cost of this strategy will be $\min(k, B + x - 1)$, and the offline optimal cost will be $\min(k, B)$. The corresponding entry in the payoff matrix $V$ is $\frac{\min(k, B+x-1)}{\min(k,B)}$, and the competitive ratio of this deterministic algorithm is $\max_{k \geq 1} \frac{\min(k, B+x-1)}{\min(k,B)}$, i.e., for the worst possible input $k$.

## 2.3 ADWORDS

We formally define the (non-relaxed) **AdWords** problem (Mehta et al., 2007): There are $m$ advertisers with budget $B$ each, and $n$ ad slots. Each ad slot $j$ arrives sequentially along with a vector $v^j = (v_0^j, v_2^j, \ldots, v_{m-1}^j)$ where $v_i^j$ is the bid of advertiser $i$ for slot $j$. Once a slot arrives the algorithm must irrevocably be allocated to an advertiser or no one at all. In the event that the ad $j$ is allocated to advertiser $i$, the algorithm collects a revenue which is the minimum of $v_i^j$ and advertiser $i$'s remaining budget. The remaining budget of advertiser $i$ is reduced accordingly. The objective of the algorithm is to maximize the total revenue over the entire sequence.

**MSVV algorithm.** Under a "small bid" assumption, which says that $\forall\, i, j : v_i^j \ll B$, the optimal online algorithm for AdWords is the following by Mehta et al. (2007). Define $s_i^j := \frac{r_i^j}{B}$ as the fraction of advertiser $i$'s budget remaining when ad $j$ arrives. Let the "scaled bid" of the slot $j$ for $i$ be $q_i^j := v_i^j(1 - e^{-s_i^j})$. The ad slot $j$ is assigned to the advertiser $i$ with the largest scaled bid $q_i^j$ (with arbitrary tie breaking). The algorithm always guarantees at least a $(1 - 1/e) \approx 0.632$ fraction of the offline optimal revenue, which is asymptotically optimal for large $n$ and $m$; smaller instances can have better (bigger) CRs.

Here, we will study the fractional version of the problem.

**Problem 2 (Fractional AdWords)** *The fractional problem is the same as in AdWords, except that the algorithm can make fractional allocations. That is, for each ad slot $j$, the algorithm chooses an assignment $(p_0^j, p_1^j, \ldots, p_{m-1}^j) \in \mathbb{R}_{\geq 0}$ where $\sum_{i=0}^{m-1} p_i^j \leq 1$. The budget of each advertiser $i$ with remaining budget $r_i^j$ is then depleted by $\min(r_i^j, p_i^j \cdot v_i^j)$ and the algorithm's revenue is increased by the same amount (summed over all $i$).*

One way to interpret the fractional AdWords as a discrete AdWords instance is to view each ad $j$ as $Q \to \infty$ infinitesimal sub-slots, and each advertiser $i$ has bid $v_i^j/Q$ for each such sub-slot. Under this view, we can derive what the optimal algorithm must look like. Suppose the algorithm chooses

the allocation $(p_0^j, p_1^j, \ldots, p_{m-1}^j)$ for ad $j$, then by considering the MSVV allocation of the "last" infinitesimal sub-slot of $j$, the following optimality condition should hold: all $i$'s where $p_i^j > 0$ must achieve the maximum $v_i^j \cdot (1 - e^{-(s_i^j - (v_i^j p_i^j / B))})$ value (among all advertisers).

Fractional relaxations may sometimes lose some combinatorial flavor of the original problems, but it is meaningful (and common) to study them in algorithm design: often they can lead to randomized algorithms or can be rounded to integral solutions. For Adwords, the fractional version retains the essential characteristics, and the optimal algorithms and hard distributions in fractional and discrete cases have a strong connection. The advantage is that the fractional relaxation leads to easy evaluation of the competitive ratio without sampling the policy, which makes the program fully differentiable, and this is very useful for our training framework.

## 3 YAOGAN

In this work, the goal is to explicitly find the minmax strategy for the row player, by co-training the row and column player networks. YaoGAN facilitates this by training both an *algorithm network* and an *adversarial network*. On a conceptual level, YaoGAN alternates between (a) training the adversarial network to produce the best response to the algorithm network (in terms of the payoff matrix, i.e., competitive ratios), and (b) training the algorithm network on the generated input. We give a brief theoretical motivation behind YaoGANs in Appendix B.

### 3.1 YAOGAN PARADIGM

We instantiate a trainable neural network representing the algorithm that starts by outputting meaningless (random) values, and our goal is to train it into a competitive algorithm. Similarly, we use some method to generate (approximately) best response inputs: This can either be an *exhaustive search* through all deterministic inputs, or can be an trainable *adversarial neural network* that starts off with outputting meaningless values and can be tuned. We focus on the more interesting, latter, case. Finally, we are given a way to evaluate the performance of the algorithm on a input. In the realm of online algorithms this corresponds to running the algorithm network, running a solver to compute the offline optimum and then reporting the competitive ratio.

**Experience array.** We maintain an *experience array* of input instances that we use to train the algorithm neural network. At each training step of the algorithm, we take some number of samples from the experience array, evaluate the algorithm's performances (i.e., the competitive ratio in our case), find the worst input among them for the current algorithm, and train the algorithm against that input.

All instances in the experience array are generated by the adversarial network at some point during the training. At every step the algorithm network is trained against the best response from some number of samples from the experience array. We periodically append new input instances to the experience array, which are chosen among inputs generated by the adversarial network and the samples from the current experience array. We append among these candidates the one that is the best response against the algorithm network.

This stabilizes the training and resolves a multitude of issues. The most apparent being that training the adversarial network to convergence takes significantly more time than one step of the algorithm training. Experience arrays amortizes the time spent during adversarial training with an equal amount of algorithmic network training. Furthermore, an empirical issue that has been hindering the training is that the adversarial network can often fail to find the best response possible. This is not at all unexpected since the question can be in general intractable. Experience arrays allow the adversarial networks to use multiple tries in finding the best counterexample. This counterexample might then be used multiple times since one often needs to train on it a few times in succession or revisit it during later training.

One explanation on why such training is effective is that hard distributions often have small support (a small number of different inputs). This is the case in Kong et al. (2018) where the handcrafted adversarial distributions are often comprised of a small number of different inputs (although their structure might be complex). Furthermore, even if all sets are of large size, one might still find a satisfactory small-set approximation that converges to the right set.

**Pseudocode.** The pseudocode of YaoGAN is given in Algorithm 1. We make a final note that our adversarial network takes $n_{\text{noise}}$-dimensional Gaussian noise as input and generates a problem input. The random noise was typically attenuated during training, but we empirically found that it helped with input space exploration. We use hyperparameters $n_{\text{batch}} = 100$, $n_{\text{noise}} = 100$, $T_{\text{alg}} = T_{\text{adv}} = 4$, $T_{\text{add}} = 100$, and $T_{\text{restart}} = 100$. The Adam (Kingma & Ba, 2014) optimizer is used with paper suggested parameters.

---

**Algorithm 1** Generic YaoGAN Training Paradigm.

---

**Require:** Differentiable evaluation function $V(\cdot, \cdot)$
**Require:** Parameters $T, T_{\text{alg}}, T_{\text{adv}}, T_{\text{add}}, T_{\text{restart}}, n_{\text{batch}}$.
  Instantiate algorithm network ALG, adversarial network ADV
  Generate $n_{\text{batch}}$ inputs using ADV, and add them to the experience array $E$.
  **For** $t = 1, \ldots, T$
    **For** $u = 1, \ldots, T_{\textbf{alg}}$
      $I \leftarrow n_{\text{batch}}$ uniformly randomly sampled inputs from $E$.
      $i^* \leftarrow \arg\max_{i \in I} V(\text{ALG}, i)$
      Update ALG using Adam($\nabla_{\text{ALG}} V(\text{ALG}, i^*)$).
    **For** $u = 1, \ldots, T_{\textbf{adv}}$
      Sample $n_{\text{batch}}$ random Gaussian tensors $z$
      Update ADV using Adam($\nabla_{\text{ADV}} V(\text{ALG}, \text{ADV}(z))$).
    **If** $t \bmod T_{\text{add}} = 0$
      Generate $n_{\text{batch}}$ inputs using ADV, sample $n_{\text{batch}}$ inputs from $E$.
      Among the above inputs, pick the worst one for ALG, append it to $E$.
    **If** $t \bmod T_{\text{restart}} = 0$
      Reinitialize ADV to produce random (untuned) problem inputs.

---

## 4 ADWORDS

In this section, we show how to apply the YaoGAN paradigm to the fractional version of the classic AdWords problem. The algorithm we derive here will apply to fixed-length inputs. In Appendix C, we present an application of the YaoGAN paradigm to derive a uniform algorithm (one that works for all input lengths) for the *ski rental* problem.

### 4.1 DEMONSTRATION OF RESULTS

In this section we demonstrate some qualitative behaviors of our training result. We fix $m = 9, n = 3$ and a common budget $B = 3$. We defer the representation details to Appendix D.1 and extra empirical results to Appendix D.2. About 10 hours of training is sufficient to achieve a competitive ratio of $\approx 0.7327$, which is within $10^{-2}$ of the theoretically best value achievable for this problem.

We start with a brief discussion of the optimal algorithm and input distribution so we can better interpret our results. At a high level, there are two intuitive aspects of the AdWords problem (in the discrete case): the *greedy strategy* and the *balance strategy*.

**Greedy strategy** allocates an ad slot to the advertiser with the highest bid for the ad when their remaining budgets are equal. This is a natural strategy to maximize revenue.

**Balance strategy** allocates an ad slot to the advertiser with the largest remaining budget when all advertisers have the same bid for the ad. This is also natural to avoid depleting the budget of any advertiser, since we may run into a future ad where only the advertiser with depleted budget has high bid for the ad. The "scaled bid" used in the MSVV algorithm (even for the fractional version) can be viewed as a careful interpolation between two strategies when the advertisers' bids and remaining budgets differ.

We first demonstrate that our trained algorithm captures the "greedy" aspect in Figure 1. For example, in the leftmost plot, we fix the remaining budgets of all advertisers at 2, the last two advertisers' bid for an ad at $0.25$ (the vertical dotted line), and vary the bid $x$ of advertiser 0 to plot the fractional allocations returned by the algorithm for different $x$ values. We can see that the algorithm correctly

allocates more to advertiser $0$ as the bid $x$ increases, and also correctly allocates roughly the most to the advertiser with the highest bid. We observe consistent behavior for other threshold values in the next 3 plots (and more comprehensive plots in the appendix).

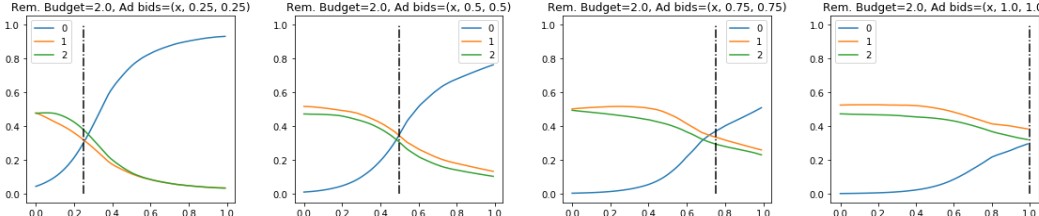

**Figure 1:** Each plot shows the fractional assignment of a new ad to the three different advertisers indexed $\{0, 1, 2\}$ while keeping the remaining budget equal across all advertisers. The x-axis shows the bid of the first advertiser and the y-axis shows the algorithm agent's fractional assignment $p$.

We demonstrate the "balance" aspect of our trained algorithm in Figure 2. In the leftmost plot, we vary the remaining budget of advertiser $0$ while fixing the other advertisers' budget at $0.9$, and all advertisers' bids are $1$. The algorithm correctly allocates more the advertiser $0$ when he/she has more remaining budget, and also correctly gives roughly the most fractional share to the advertiser with largest remaining budget. Again, the observation is consistent in the other plots (and more in the appendix). We note that different from the discrete case where one should expect a sharp transition

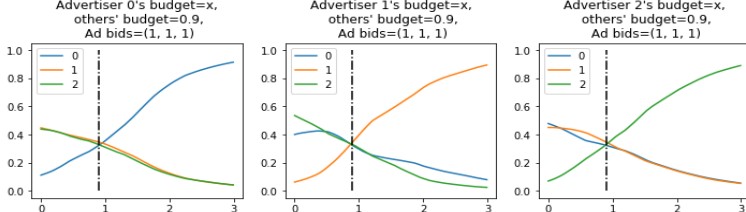

**Figure 2:** Each plot shows the trained agent's allocation of a new ad to three different advertisers indexed $\{0, 1, 2\}$. The ad has a common bid of $1$ from all advertisers. In the $i^{th}$ plot ($i \in \{0, 1, 2\}$) the x-axis denotes the budget of advertiser $i$, while other advertisers have a common remaining budget. The y-axis shows the algorithm agent's fractional assignment $p$.

of allocation around the threshold, the smooth curves observed in our plots actually agree with the optimal algorithm in the fractional case. For a concrete example (see Figure 3), we consider the case where the remaining budgets of the advertisers are all fixed at $1$, and advertisers have bids $(x, 0.5, 0)$ for varying $x$ values. These parameters are chosen for simpler calculation. Let's consider the optimal algorithm's allocation in this case. From our discussion of the optimal fractional algorithm (immediately following the problem definition of fractional AdWords), we know that the optimal algorithm will split the ad between advertiser $0$ and $1$ (as advertiser $2$ has bid $0$), and if the fraction allocated to advertiser $0$ is $t \in [0, 1]$, and $1 - t$ to advertiser $1$, we must have

$$x \cdot (1 - e^{-\frac{1-t}{3}}) = 0.5 \cdot (1 - e^{-\frac{1-(1-t)}{3}})$$

If we use the approximation of $e^{-x} \approx 1 - x$ for small $x$, the rough calculation gives $t$ as a function of $x$ where $t = \frac{2x}{2x+1}$. We plot this as the dotted (red) curve in Figure 3, and we observe that the fractional allocation to advertiser $0$ (as a function of $x$) by our trained algorithm is fairly consistent with the optimal.

On the adversary side, in the specific case of $3$ advertisers and $9$ ad slots, an adversarial input distribution (i.e. a minmax column strategy in the game theoretical discussion of Section 2.1) is the uniform distribution over the $6$ matrices which are the column permutations of the transpose of the following matrix, where ads arrive from top to bottom (left-to-right in the transposed matrix below).

$$\begin{bmatrix} 1 & 1 & 1 & 1 & 1 & 1 & 1 & 1 & 1 \\ 1 & 1 & 1 & 1 & 1 & 1 & 0 & 0 & 0 \\ 1 & 1 & 1 & 0 & 0 & 0 & 0 & 0 & 0 \end{bmatrix}$$

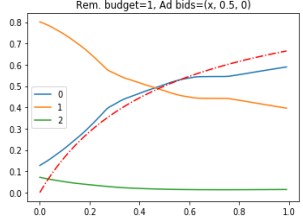

**Figure 3:** The plot shows the trained agent's allocation of a new ad with bids $(x, 0.5, 0)$, where $x$ is the value on the x-axis and y-axis denotes the fractional assignment $p$. All advertisers have a common budget of 1. The dotted (red) curve depicts the approximate behavior of the optimal algorithm for bidder 0.

In the previous work by Kong et al. (2018), they showed that a mixture of an adversarial input distribution and a high-entropy (i.e. noisy) distribution gives a sufficiently good (pre-determined) training set. We plot a set of random samples from the experience array at the end of our training in Figure 8 in Appendix D.2. It is clear that without any expert input, the self-generated input instances during training capture enough structure of such an expert crafted mixture. We find this highly non-trivial.

## 5 CONCLUSION AND FUTURE DIRECTIONS

In this work we explore the idea of using adversarial training for learning optimal algorithms from first principles. We propose the YaoGAN technique and use it to recover several near-optimal online algorithms without using any problem-specific domain knowledge about the problem, courtesy of adversarial training. Our results suggest YaoGAN is effective in learning algorithms that are robust to worst-case inputs. We also show an important limitation of our approach: The training is as effective as the search for best response inputs. When this search can be effectively performed the trained algorithm seem to be near optimal; if the search is approximate, the algorithms are also approximate.

The ultimate goal of our investigation is to gain crucial insight into interesting open problems in previously unexplored ways. While this goal is currently out of reach, we hope our research opens up such a path. On a less ambitious note, our future directions include (1) to use adversarial training to develop algorithm independent of problem size for hard problem such as AdWords, (2) train offline algorithms, and (3) solve general online packing/covering problems as an immediate extension of our techniques for AdWords.

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

## A  ONLINE ALGORITHM

In this section, we provide more background on the essential notions of online combinatorial optimization for the ML audience.

Online algorithm is a very different computational model from the classical (offline) algorithm design, and arises broadly in practice in situations where one has to make decisions in real time without knowing the future.

In the online context, the input is revealed sequentially instead of all at once, and the algorithm needs to make irrevocable decisions during the process using the input revealed so far. The difficulty comes from making irrevocable decisions without knowing future input, while the current decision will have implications in the future. Take AdWords as an example: the search engine can sell an ad slot (i.e., user impression) each time some user searches for some query. Advertisers specify their budgets (i.e., total amount of money they are willing to spend on all ads) at the beginning of the episode and then place bids each time a new ad slot arrives. The search engine has to decide in real time how to allocate the ad slot to advertisers given their bids and budgets. The goal for the

search engine is to maximize its revenue over the entire input sequence. Once we display the ad from an advertiser, there is no way to change the decision later, and once the budget of an advertiser is depleted, the search engine cannot sell ads to the advertiser any more. See Section 2.3 for a formal problem definition.

**Inherent difficulties in online optimization.** Consider the instance with two bidders $A, B$ both having budget 1, and two ad slots arriving sequentially. Suppose $A$ and $B$ both bid 1 for the first ad. To whom do we allocate the ad? With out loss of generality, suppose we give the first ad to $A$, and collect payment of 1 from $A$. Now suppose the second ad slot comes, and only $A$ bids 1 for it (and $B$ bids 0). We cannot collect any more revenue from $A$ as $A$'s budget is already spent on the first ad, and cannot collect any revenue from $B$ since $B$ values the second ad at 0. In this case the total revenue we get is 1. However, hypothetically, if we knew the entire input sequence before making any decision, we could have assigned the ads optimally (the first one to $B$ and the second to $A$), getting a total revenue of 2. This revenue is called the *offline optimal*, i.e., how well we could have done in retrospect.

**Competitive ratio (CR).** The main objective for online optimization is to design algorithm with good CR. This is in contrast to (offline) algorithm design where the emphasis is mostly on efficiency. The CR of an algorithm with respect to a particular input is the ratio between the result of the algorithm on this input over the offline optimal of the input. In the above example, we get a competitive ratio of $1/2$ on that particular instance.

**Adversarial model.** There are various models specifying what assumptions we have on inputs (i.e., on what inputs we want the algorithm to achieve good CR), e.g. i.i.d, random order, or adversarial. In this paper we assume the, most general, *adversarial model*. In it, we want the algorithm to achieve good CR against all possible inputs, and we refer to the CR of an algorithm as the worst CR the algorithm gets on any input. This model achieves the highest level of robustness since it does not put any assumptions on the input; making it a meaningful model to consider in mean real-world settings. For example, in online advertising, the ad slots supply (i.e. user searches) can change drastically when some events go viral. Moreover, advertisers spend a great amount of resources to reverse-engineer the search engine's algorithm and come up with sophisticated bidding strategies in response. So it is of great interest to design algorithm that performs well no matter how the inputs look like.

## B    NO REGRET DYNAMICS AND CONVERGENCE TO THE MINMAX

There is a rich line of work on no regret dynamics (Freund & Schapire, 1996; Arora et al., 2012) which lies at the center of various fields including game theory, online learning and optimization. In the context of two-player zero-sum games, this gives a method to approximately solve the value of the game without knowing the optimal column player strategy (i.e., the adversarial input distribution) in advance. We extend the notation to use $V(\boldsymbol{p}, \boldsymbol{q})$ to denote $\boldsymbol{p}^T V \boldsymbol{q}$.

More specifically, we can let the algorithm player maintain a randomized algorithm $\boldsymbol{p}^t$ over iterations $t = 1, \ldots, T$. In each step the adversary plays the best response $\mathcal{I}^t$ (i.e. the worst-case input against the algorithm at that iteration). The algorithm $\boldsymbol{p}^{t+1}$ is then computed from $\boldsymbol{p}^t$ and $\mathcal{I}^t$ using a specific no-regret dynamic such as Hedge (Freund & Schapire, 1996). Define

$$\tilde{\boldsymbol{p}} = \arg \min_{\boldsymbol{p}^t} V(\boldsymbol{p}^t, \mathcal{I}^t), \qquad \tilde{\boldsymbol{q}} = \frac{1}{T} \sum_t \mathcal{I}^t.$$

Classic results give us that the payoff of the pair $\tilde{\boldsymbol{p}}, \tilde{\boldsymbol{q}}$ converges to the optimal minmax value for sufficiently large $T$.

YaoGAN loosely follows this approach. It maintains a randomized algorithm $\boldsymbol{p}^t$ and searches for the best response $\mathcal{I}^t$. However, it does not follow the no-regret dynamics of prior algorithms, but rather uses simple gradient descent to optimize the payoff of the algorithm on $\mathcal{I}^t$. We avoid going into formal details since we use the theoretical guarantees mostly to draw insights.

## C    SKI RENTAL

In this section, we derive a uniform algorithm for the classic ski rental problem (defined in Section 2.2).

A randomized algorithm for the ski rental problem is represented by a (marginal) probability distribution $(p_1, p_2, \ldots, p_N)$, with the semantics that for each $x \in \{1, 2, \ldots, N\}$, the value $p_x$ denotes the probability that the we rent skis for the first (up to) $x-1$ days and buy skis on day $x$, if the season extends that far. For reference, in the most interesting case $N \to \infty$ we note that the optimal choices are given by $p_i = (\frac{B-1}{B})^{B-i} \frac{c}{B}$ for $i = 1, \ldots, B$, where $c = \frac{1}{1-(1-1/B)^B}$ so these probabilities sum to 1. It has a competitive ratio of $\frac{1}{1-(1-1/B)^B}$, which tends to $\frac{e}{e-1} \approx 1.582$ as $B \to \infty$ (Karlin et al., 1986).

Our goal here is to learn an algorithm that works for *all* values of $B$ and $N \to \infty$. This is out of reach for traditional optimization methods for a number of reasons, the most obvious one being the infinite number of variables one needs to optimize over.

For the ski rental problem, we apply a simpler variant of the YaoGAN paradigm. Specifically, we train a deep neural network to represent the algorithm, but for the adversary, we don't resort to deep neural networks yet. This decouples the algorithmic from the adversarial training and showcases that best-response training is effective. Note that one can still use the full YaoGAN paradigm if one chooses do to so. However, this would be an "overkill" since finding the best response in this case (unlike the AdWords case) is a simple problem where essentially any reasonable method is going to be effective.

As $B$ and $N$ can be (asymptotically) large, we need to work with a continuous space, where every parameter is essentially normalized by $N$. In this setting, time goes over the range $[0, 1]$ indicating what fraction of the time horizon has passed. At time $\alpha \in [0, 1]$, if we are still skiing, the cost of renting from the beginning of time up to this point is captured by $\alpha$. The cost of buying is captured by a real number $\beta \in [0, 1]$, which is the fraction of the time one needs to go skiing in order to make buying the skis worthwhile. If we stop skiing at point $\alpha$ of the continuous timeline, the optimal offline cost is $\min(\alpha, \beta)$.

**Applying the YaoGAN paradigm.** On the algorithm side, we train a deep neural network that takes $\alpha, \beta \in [0, 1]$ as inputs, computes the CDF of a probability distribution $p_\beta$, and outputs the value of the CDF at $\alpha$. As for the architecture of the algorithm neural network, we use 50 Gaussian kernels $N(x_i, \sigma)$ for $i = 1, \ldots, 50$ with fixed means (i.e., $x_i$'s equally spaced over $[0, 1]$) and standard deviation $\sigma = 2/50$, and take a weighted average of these Gaussian kernels as the distributions learned by our neural network. In particular, given any $(\alpha, \beta)$ as input, the neural network uses $\beta$ to derive weights $w_i$'s over the Gaussian kernels (where $\sum_i w_i = 1$), and returns the weighted sum of the CDF of these fixed Gaussian kernels at point $\alpha$ as the output of the algorithm. We use a fairly generic neural network structure to learn the kernel weights: the network takes $(\alpha, \beta)$ as inputs and first applies 4 hidden ReLU-activated layers with 256 neurons. The last hidden layer, the one that outputs $w$, is a softmax-activated. Adam optimizer Kingma & Ba (2014) with the default parameters is used to train the network

On the adversary side, in each training iteration, we randomly sample a small number of values for $\beta$, and for each sampled $\beta$ we use an $\epsilon$-net over $[0, 1]$ as values of $\alpha$ with $\epsilon$ chosen to be 0.01. We approximately evaluate the competitive ratios (up to certain precision) of the current algorithm for these $(\alpha, \beta)$ instances, and train the algorithm neural network with the instance where it gets the worst competitive ratio. The approximation is due to not really taking a integral over the learned cdf but rather a step function approximation of it.

We demonstrate the trained algorithm in Figure 4 by instantiating it at various values of $B$ and $N$. The result is very close to the optimal strategy in all cases, and indicates that our theoretical insights from Section B largely extends to more general settings. We also note here that the competitive ratio achieved by this algorithm converges approximately to 1.59 within a few minutes of training, about 0.01 worse than the optimal value (1.582).

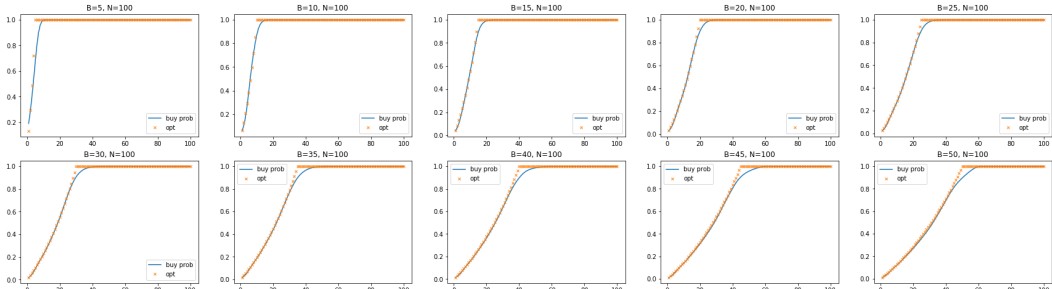

**Figure 4:** The learnt algorithm instantiated at various $(B, N)$ values as CDF of buying over time (blue curve). The x-axis denotes the number of days the algorithm has seen so far and the y-axis denotes the buying probability. The blue curve is the trained agent's output, while the orange correspond to the theoretical optimal strategy.

# D    ADWORDS

## D.1    ALGORITHM AND ADVERSARY REPRESENTATIONS

In this work we train an algorithm with a fixed number of advertisers $m = 3$. Although this is a rather limited setting, it captures the essential structure of the problem, and we can observe very non-trivial behavior from both the algorithm side and the input side. There does exist a representation for a uniform algorithm, i.e., one that works for arbitrary number of $m$ (e.g. see Kong et al. (2018)), and we leave it for future work to extend our framework in that set-up.

Upon the arrival of an ad, the neural network for the algorithm takes as input a vector containing the bids of the 3 advertisers for the current ad and their remaining budgets. The neural network outputs a probability distribution $(p_0, \ldots, p_3)$ indicating the fractional allocation of the ad (with $p_3$ being the fraction not given to anyone). The network itself is composed of 4 hidden and ReLU-activated layers with 500 neurons succeeded by a softmax output layer. The training environment feeds each input instance to the algorithm sequentially, and updates the total revenue and advertisers' remaining budgets along the way with the algorithm outputs.

On the adversary side, we consider inputs with a fixed number of ad slots $n = 9$, and initial budget $B = 3$ (both can be changed to arbitrary values if needed). The network itself takes $n_{\text{noise}}$ Gaussian variables, acts upon them using 4 hidden ReLU-activated layers of 200 neurons each with batch normalization and outputs a $m \times n$ sigmoid-activated matrix (with outputs in $[0, 1]$).

It is important to note that in this problem (as opposed to the ski-rental problem) there is no easy exhaustive strategy for the adversary side, even with discretization of the weights on the edges (as there are 27 numbers in an input graph).

## D.2    ADDITIONAL PLOTS FOR ADWORDS

**Training convergence**    In Figure 5, the green curve tracks the estimated quality of our trained algorithms saved at different snapshots along training (every 100 training iterations). Note these are estimated CR, since the true CR of an algorithm is the worst CR it gets over all possible inputs, but we cannot enumerate over all (infinitely many) possible inputs. Instead, we estimate the CR of each algorithm snapshot using the experience array (generated by the adversarial network) as a benchmark. That is, after training, for each algorithm snapshot we evaluate its CR against all instances in the entire experience array (not the partial experience array at the time of the snapshot), and plot the worst CR it achieves. We expect the green curve to go up, which suggests our algorithm gets better over training. The black dotted line is the CR of the theoretically optimal algorithm, and we see the green curve approaches it. The green curve actually goes slightly above it towards the end, which is due to the fact our CRs are only estimations, so it can be higher than the true CR for our algorithm if the experience array doesn't contain the absolute worst possible input instance for the algorithm.

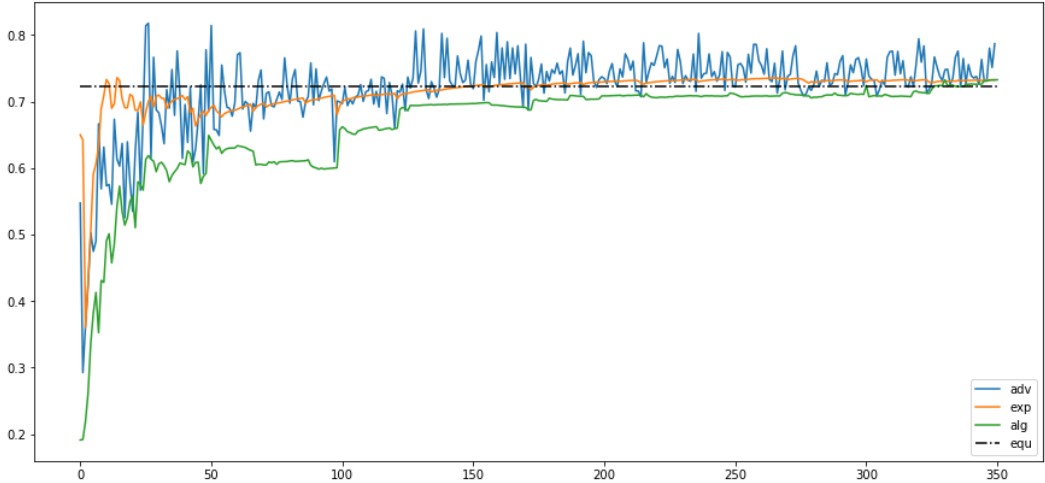

**Figure 5:** Competitive ratio at snapshots taken just before the adversarial network is restarted. The x-axis represents the snapshot number and the y-axis shows the competitive ratios. The blue (adv) line represents the CR of the fully trained adversarial network before its restart. The orange (exp) line at snapshot $i$ represents the CR of the best response from the experience array at snapshot $i$ versus the trained algorithm at snapshot $i$. The green (alg) line at snapshot $i$ represents the CR of the best response from the entire experience array (from the entire training) versus the trained algorithm at snapshot $i$.

Because we can only estimate the CRs, there is the issue of potential collusion. That is, the algorithm network may only find very poor algorithm, but the adversarial network also only generates very easy instances, so the algorithm performs well against generated instances, and thus we get high estimated CRs. In our result, the fact that the estimated CR curve converges to the theoretical optimal CR eases this worry of potential collusion to a large extent, as the two networks have no knowledge about this optimal value, thus highly unlikely to generate algorithms and inputs that happen to give that particular estimated CR.

Note this is an intrinsic issue common to problems of such nature. For example, in the case of chess, people estimate the strength of a board configuration or a move not by any ground truth but by heuristic search and self-playing. However, this can never eliminate the possibility that an estimated good move can lead to a loss if the real opponent is much more powerful than the one used in training, and finds a very good counter-move. Similar situation holds in the context of adversarial training (Madry et al., 2018). In all these cases, all people can do is to try to make the adversary side as good as possible during training, so the estimations won't be too far off.

**Algorithm behavior** In Figure 6 and Figure 7 we examine the behavior of the learned algorithm upon the arrival of an ad slot. See Section 4 for the qualitative behavior of the theoretically optimal algorithm. In Figure 6, we vary the bids of the advertisers for the ad slot while keeping their remaing budgets the same. Our algorithm clearly demonstrates the greedy behavior (which is the correct strategy in this scenario) of the optimal algorithm. In Figure 7, we fix the bids of all advertisers for the ad slot, and vary their remaining budgets. Our algorithm clearly demonstrates the balance behavior (again the correct strategy in this scenario) of the optimal algorithm. In both figures, the vertical axis is the fractional allocation of the ad slot to the advertisers.

**Adversarial distribution** We now offer some intuition to interpret Figure 8. For AdWords (also holds for fractional AdWords), there is a known adversarial distribution over input instances that certifies no algorithm can get better than the optimal CR. Qualitatively, a hard instance follows the pattern that: 1) many advertisers have high bids for the earlier ad slots, and 2) fewer advertisers have high bids for the ad slots arriving later. For such input, if the algorithm is not smart at choosing the allocation for the earlier ads, it may deplete the budget of some advertisers, and run into the situation where advertisers with high bids for the later ads already have their budget depleted, while those with remaining budget don't value highly of the later ads. If we restrict ourselves to the instances over 3

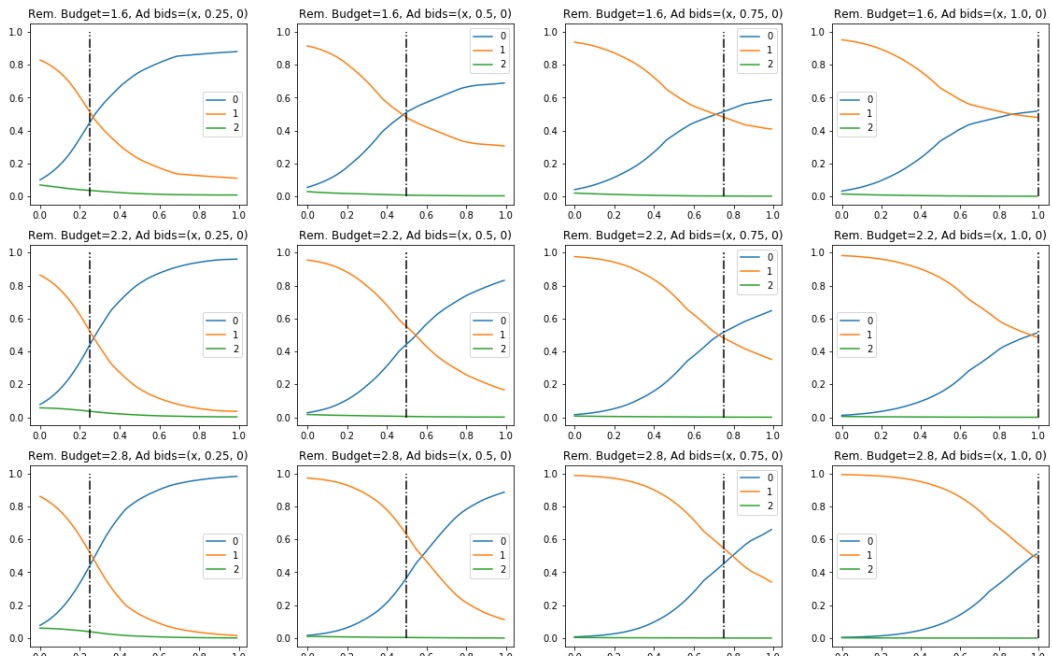

**Figure 6:** Each plot shows the fractional assignment of a new ad to the three different advertisers indexed $\{0, 1, 2\}$ while keeping the remaining budget equal across all advertisers. The bids of the incoming ad slot are $(x, c, 0)$, where $x$ changes along the x-axis and $c$ is common for every plot in the same column. The y-axis shows the fractional assignments $p$. Each row of plots features advertisers with a common remaining budget.

advertisers and 9 ad slots, a concrete instance is given as the triangular matrix at the end of Section 4 (copied below).

$$\begin{bmatrix} 1 & 1 & 1 & 1 & 1 & 1 & 1 & 1 & 1 \\ 1 & 1 & 1 & 1 & 1 & 1 & 0 & 0 & 0 \\ 1 & 1 & 1 & 0 & 0 & 0 & 0 & 0 & 0 \end{bmatrix}$$

The 9 ads arrive sequentially from left to right, the $(i, j)$-th entry is the bid of advertiser $i$ for ad $j$, and all advertisers have a budget of 3. However, this instance alone is not enough, since it can be exploited by a naive strategy that always allocates ads to advertiser 3 (until 3 runs out of budget), then advertiser 2, then advertiser 1. This naive strategy achieves the same as the optimal offline revenue of 9, and thus a competitive ratio of 1 on this one instance. The adversarial distribution is over permutations of the advertisers' names (i.e. rows of the matrix), so naive strategy mentioned before no longer works well.

Figure 8 plots samples of the "best responses" generated during training. Note these matrices are transposed. In each matrix, ad slots are rows with the top row arriving first, brighter entries mean higher bids (between 0 and 1), and the matrices are ordered by when they are used to train the algorithm. Although we don't have any quantitative measure, it is fairly clear that the pattern discussed in the previous paragraph emerges as the training evolves. In particular, most of the instances generated later in the training clearly demonstrate the pattern that earlier ads (i.e. top rows) have more advertisers bidding highly (i.e. brighter cells), and later ads have less. Furthermore, we can also see that the adversarial network learns to permute the advertisers (i.e. columns) fairly well. It is very interesting that the adversarial neural network can come up with such pattern from scratch via interactively playing against the algorithm network.

The generated instances are noisy compared to the adversarial distribution in theory (i.e. the triangular matrix and its permutations). This is also the desired behavior as a good training set is expected to be high-entropy (see discussion at end of Section 4).

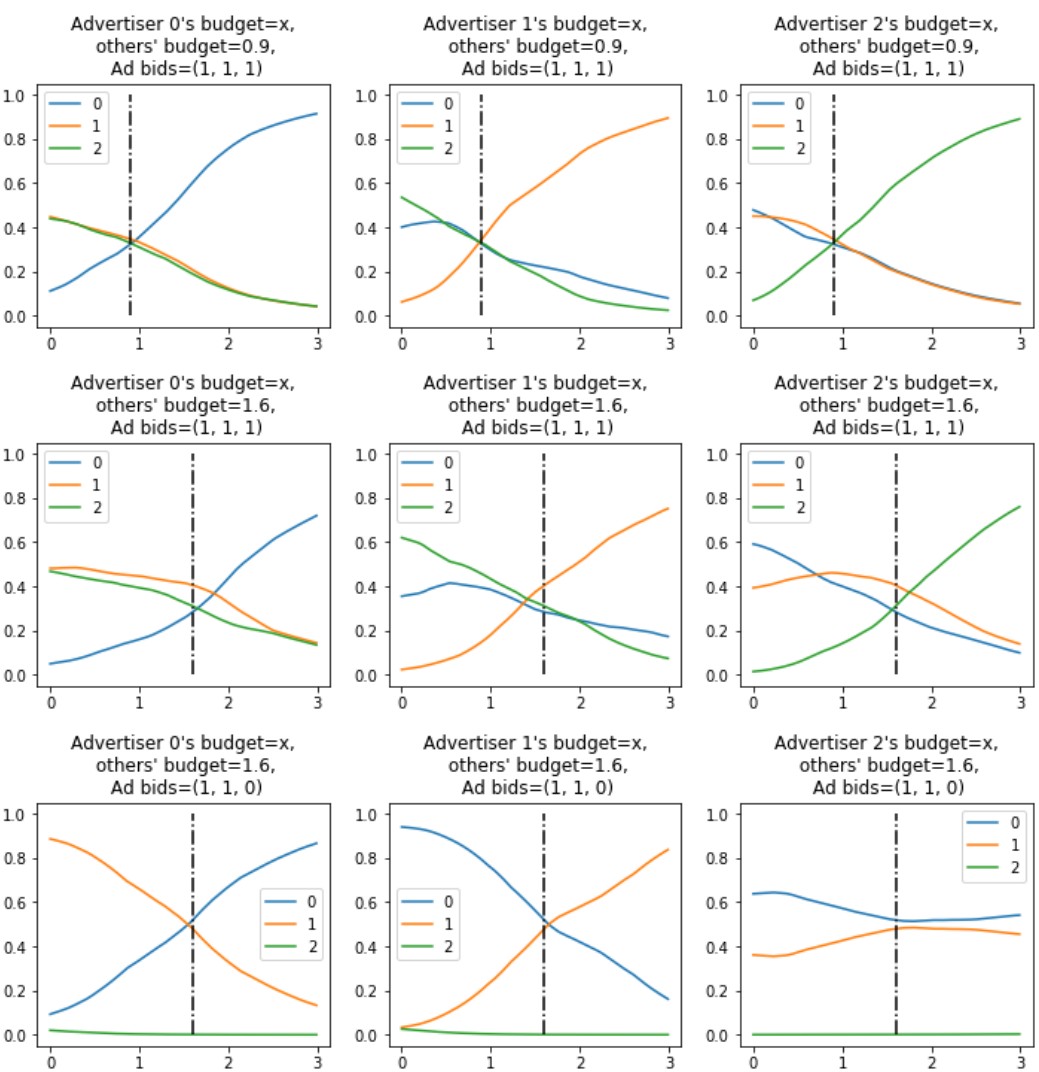

**Figure 7:** Each plot shows the fractional assignment of a new adword to the three different advertisers indexed $\{0, 1, 2\}$. The ad slot has a common bid of 1 to all advertisers. The remaining budgets change for one advertiser along the $x$ axis while staying the same for all other advertisers. The y-axis denotes the fractional assignment $p$.

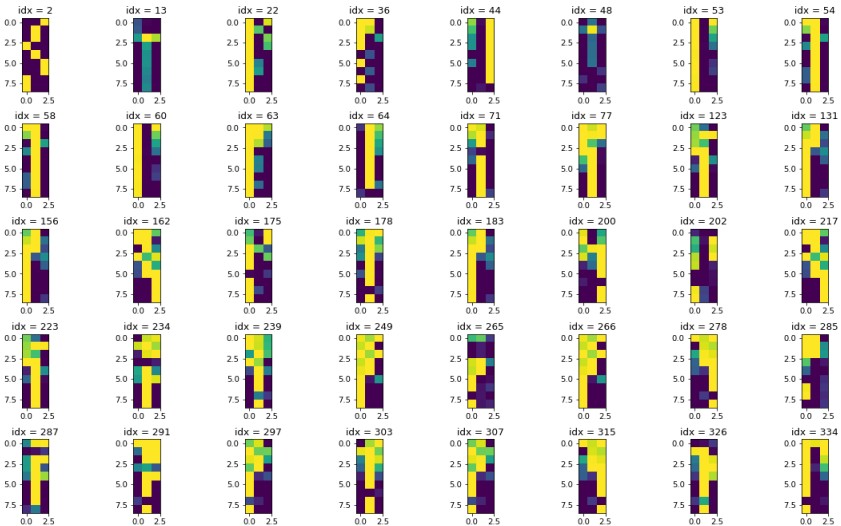

**Figure 8:** A set of random samples from the experience array after training. The $j^{th}$ row and $i^{th}$ column represent the bid of advertiser $i$ for ad slot $j$. Darker color represents a bid close to $0$, while brighter color represents a bid close to $1$.

