# OpenReview forum: "YaoGAN: Learning Worst-case Competitive Algorithms from Self-generated Inputs"
_ICLR.cc/2020/Conference — Reject_

### Official Review · AnonReviewer1 · 2019-10-11
**Official Blind Review #1**

**Rating:** 3

**Review:**

This papers tackles the following question. Is it possible to learn the "most" complex instance of a class of (combinatorial) problem while finding (or recovering) algorithms with strong minimax rate.

This is very interesting and clearly a nice line of work (in theory though).

The techniques used rely on GANs since it can be shown that finding the best (random) algorithm and the worst (deterministic) instance is equivalent to finding the worst random instance against the best deterministic algorithm. This is actually a direct consequence of any minmax theorem in game theory; the authors decided to credit that result to Yao (I tend to *strongly* disagree with that point as, even if he stated this fact in CS, this result was quite standard several decades before him - anyway.).

Then this idea is evaluated in two examples. A toy problem (the ski rental) and a more or less concrete ones (adwords pb of Mehta). This is the major disappointment in the paper. The basic idea is very interesting, but I would have expect more interesting use cases as teased by the first sentence of the abstract "find algorithms with strong worst-case guarantees for online combinatorial optimization problems".

So at the end, I am a bit puzzled. I really like the idea, but I have the feeling that this technique should have been developed for more complicated setting. Or maybe it is actually not working on more difficult combinatorial problem (and this is hidden in the paper). I believe that this paper is thus not in its final form and could be largely improved.







**Experience Assessment:**

I have read many papers in this area.

**Review Assessment: Checking Correctness Of Derivations And Theory:**

I assessed the sensibility of the derivations and theory.

**Review Assessment: Checking Correctness Of Experiments:**

I assessed the sensibility of the experiments.

**Review Assessment: Thoroughness In Paper Reading:**

I read the paper thoroughly.

---

> ### Author Response · Authors · 2019-11-09
> **Response to reviewer #1**
>
>
> Thank you for your review. Please also see our high-level clarification above which we believe can help in better interpretation of our contribution. Some specific responses below:
>
> Reviewer #1 is absolutely right -- we don’t know yet how to scale this to more difficult combinatorial problems.  But let’s clarify that statement a bit more:
>
> The ski-rental problem is often the first problem studied when teaching online algorithms, but it is certainly far from a “toy problem” when we wish to learn an algorithm from scratch. We apologize if we painted an incorrect picture by calling it a “simple example” and a “staple introductory problem”.  It is easy to describe in that it has a single hidden parameter (the length of the ski season) and a single revealed parameter (the cost of buying).  It is a staple introductory problem because it is elegant and illuminates the essential difficulties in designing online algorithms: there is a nearly-trivial factor-2 competitive algorithm (rent until you’ve spent $B, then buy, so even if the ski season ends the next day, you’ve not spent more than twice the least possible amount), but the 1-1/e competitive ratio algorithm is quite creative and subtle, and serves as an introduction to the richness of the field of online algorithms.   In fact, the Karlin et al. (1986) paper also introduced the notion of competitive analysis of online algorithms, and  is probably the most-cited paper in this field.  In some sense, this poses us the ideal challenge: can ML approaches discover creative and subtle “solutions” (in our case, an algorithm)?
>
> On a more technical note, please note that our “machinery” of solving the two-player game is needed to discover an algorithm for the ski rental problem: if we don’t allow the players to alternate and reach an equilibrium, for any fixed distribution on the ski rental instances (B, K), there is a deterministic algorithm that is optimal (among all online algorithms), and the worst-case performance of that (or any) deterministic algorithm is *provably* limited by a factor of 2 (i.e., there exists some distribution on instances where it will fail badly). Also refer to our discussion on this in the high-level clarification at the top.
>
> The AdWords problem considered is actually a difficult combinatorial problem, and is an archetypal online combinatorial optimization problem that captures the class of problems solvable by one of the most powerful techniques in this area -- primal-dual algorithms, which have led to the state-of-the-art approximation algorithms for numerous hard online (and offline) optimization problems.  In particular, it generalizes bipartite matching, historically one of the most significant combinatorial optimization problems (led to the development of the classic Hungarian method, see https://en.wikipedia.org/wiki/Hungarian_algorithm).
>
> We did water down our ambition in a few ways:
>
> Instead of producing an algorithm that works for inputs of all sizes, we focus on the case of 9x3 (three advertisers, nine slots) -- a fixed finite size!  This choice was arrived at based on the following criteria: what can we learn in a few hours of computation that’s still *well beyond* what can be achieved through exhaustive search (for an algorithm).  Think of our task roughly as learning to play a very hard game on a 9x3 board -- we would, of course, love to learn how to play the same game on arbitrary size boards, but the fact is that the game is mighty hard even at this “board size” (since in each round, one player plays a 0-1 assignment to each cell in the board, and the other player picks a subset of the columns, in fact a weight vector on the columns).
> Instead of producing an algorithm that works for the 0-1 version of the problem, we produce an algorithm that works for the fractional version of the problem.  This is, once again, motivated by making something work with modest amount of computation.  Our explorations indicated that producing an algorithm for the 0-1 version needs reinforcement learning, and producing an algorithm that works on all 9x3 instances using this approach would still take several days of computation.
>
>
> On calling it Yao’s Principle: as Reviewer #1 correctly noted, this is an application of the classic von Neumann minimax principle to the “game” between an “algorithm player” and an “input player”.  We call it Yao’s principle primarily in accordance with tradition in theoretical CS (see https://en.wikipedia.org/wiki/Yao%27s_principle and also https://blog.computationalcomplexity.org/2006/10/favorite-theorems-yao-principle.html, where it is noted that “Yao observed [the result]” and commentators note that it’s called Yao’s principle because this observation has significant consequences for many central problems in TCS).  We are happy to add text to reflect this.

---

### Official Review · AnonReviewer2 · 2019-10-15
**Official Blind Review #2**

**Rating:** 3

**Review:**

This paper introduces a new approach to solve optimization problems without relying on any human-provided data beyond the specification of the optimization problem itself. The approach is inspired by the two-player zero-sum game paradigm and follow a generative adversarial network (GAN) setting. One network is trained to output the optimal behavior for a given problem, while the other is trained to output difficult instances of the given problem. These two networks are trained simultaneously and compete against the other until some equilibrium is achieved.
This approach is tested on two small problems for which the optimal behavior is known and seems to perform near theoretical optimality.

I weakly reject this paper because although the approach is indeed interesting, the paper is lacking some structure, as described below:

- The paper clearly mentions that no optimization of the training setup or the hyperparameters has been done because the authors are not interested in extending ML techniques. However, hyperparameter searching is not extending any ML technique, it is just an approach to find a good training configuration and show robustness in different hyperparameters settings. It is thus unclear if the approach is robust against different hyperparameter settings.
- Very little details (apart from the optimization algorithm) are given regarding the architecture used (types of input, output, neural units, activation functions, number of hidden layers, loss function, etc...), which makes it very hard to reproduce this approach.
- Section 1.1 presents results with too many details without introducing the problem. I would suggest the authors to either introduce the two problems earlier or to simply say that near-optimal results are achieved, without giving detailed results, because it is very hard to understand them without any introduction of the task being achieved.
- One task is presented in Section 2 "Preliminaries" while the other task is presented in Section 4 "AdWords". It is hard to follow the flow of ideas present in the paper when similar things are not together. I would suggest restructuring the paper into a more classical structure such as: <intro without detailed results - previous work & problematic - approach taken with more details for reproducibility - description of the two tasks - description of experiments with more details for reproducibility - results - conclusion>.
- The paper mentions the MSVV algorithm twice but no reference or explanation is provided. It is very hard to understand sentences referring to this.
- This work only considers problems for which the optimal input distribution is known, but is motivated by the fact that it could be applied to problems for which the optimal distribution is unknown and thus being able to discover new algorithms. It is hard to support this motivation when no experiments are done in its favor.
- No comparison has been made between their approach and other previous approaches. We only know that the proposed approach finds near-optimal solutions with a difference of 0.01 competitive ratio. It is thus very hard to know if this new approach brings any improvement to previous work.

Below are a few things that were not considered to make a decision, but are only details that would make the paper slightly better:
- typo at the beginning of section 3.1: missing 'be' in  "This can either *be* by an ..."
- typo at the beginning os section 4:  missing 'be' in "... the algorithm must irrevocably *be* allocated to ..."
- Axis' names to the different plots in the Figures would help understand them better. Also, the description of some figures could benefit more details that could be taken off from the text.


**Experience Assessment:**

I do not know much about this area.

**Review Assessment: Checking Correctness Of Derivations And Theory:**

I did not assess the derivations or theory.

**Review Assessment: Checking Correctness Of Experiments:**

I assessed the sensibility of the experiments.

**Review Assessment: Thoroughness In Paper Reading:**

I read the paper thoroughly.

---

> ### Author Response · Authors · 2019-11-09
> **Response to reviewer #2, part 2/2**
>
>
> -- “This work only considers problems for which the optimal input distribution is known, but is motivated by the fact that it could be applied to problems for which the optimal distribution is unknown and thus being able to discover new algorithms. It is hard to support this motivation when no experiments are done in its favor.”
>
>
> The long-term agenda / research program is indeed two-fold:
> 1. Investigate whether known optimal worst-case algorithms can be reproduced without any domain knowledge (i.e., “Can ML learn Algorithms”). This is the case in which the optimal distribution of inputs is also known.
> 2. Discover new/better worst-case algorithms for problems with the aid of ML, when neither a good algorithms or input distribution is known.
>
> #2 is a long-term goal, and not tackled in this paper, but we believe #1 (tackled in this paper) is itself of strong interest (and difficult) -- would ML be able to discover the same “pen-and-paper” algorithms that computer scientists invented?  The problems we study (ski-rental and Adwords) fall into the first category of problems. Note that the algorithms in the two cases are very different in structure.
>
> Further, please note that even though the optimal distribution of input is known in these two problems, we do not use it at all in training. Indeed, this is the main point of this paper -- the previous work of Kong et al. used these distributions to train the algorithm network (and hence that technique still needed the prior theoretical “pen-and-paper” work), while this work starts with ZERO knowledge. We follow this approach even in case #1 when the optimal input distribution is known exactly because we have the ultimate goal #2 in mind, that is, we want to design a framework that can eventually also work without knowledge of optimal input distribution (but that goal is outside the scope of this paper).
>
>
> -- “No comparison has been made between their approach and other previous approaches. We only know that the proposed approach finds near-optimal solutions with a difference of 0.01 competitive ratio. It is thus very hard to know if this new approach brings any improvement to previous work.”
> We believe there is some misunderstanding here as to the contribution. As such, there are no previous approaches to “learn algorithms” (besides Kong et al.). To be more explicit (in case we didn’t understand the comment), previous work for algorithmic problems could fall into a few buckets:
>
> (1) The original algorithms papers which found optimal worst case algorithms [Karlin et al. 1986, Mehta et al. 2007]. These give the analytical benchmarks. E.g., [Mehta et al. 2007] proposes the algorithm to solve Adwords, and proves that it achieves the optimal CR of 1-1/e ~ 0.63 (i.e., no matter what the online input sequence is, you get >= 1-1/e of the optimal solution in hindsight if you knew the instance offline). Thus the difference of 0.01 CR is a direct comparison to that work.
>
> (2) One may imagine there could be some kind of optimization (IP / LP) technique or some ML technique to solve specific instances of the problem (a specific instance of Adwords e.g.). But this is in fact not a feasible possibility, for two reasons: (a) Our problems are online problems where the full instance itself is not known in advance, and (b) we are looking for worst case competitive algorithms, i.e., a policy that does well no matter how the instance unfolds in the future. Thus there can not be previous work to compare in such a bucket.
>
> (3) Kong et al., 2018 is the closest previous work since it shows how to learn algorithms in the online setting. As mentioned above, the critical difference is that our paper learns the algorithms without any prior knowledge of the worst input distribution, but evolves both the distribution and the algorithm jointly (with some parallels to GANs, AlphaZero, self-play, etc. as we have stated). Quantitatively, the CR results are equally good; our main objective is to see if the learned algorithm is close in policy to the theoretical algorithm, and whether we are reasonably close to the optimal CR.

---

> ### Author Response · Authors · 2019-11-09
> **Response to reviewer #2, part 1/2**
>
>
> Thank you for your review. Please also see our high-level clarification above which we believe can help in better interpretation of our contribution. Some specific responses below:
>
> ** Addressing comments on the write-up:
>
> Thanks, these help improve the presentation greatly (we realize we wrote the exposition more  from a theoretical view and missed important ML details).
>
> Details on architecture: Agreed, and thanks. We have added some details on the specific network architectures to Appendix C (for ski rental) and Appendix D (for AdWords).
>
>
> New suggested structure and related suggestions: These are nice suggestions and explain why the structure was confusing. We’ve worked on these to come close to the suggested structure.
>
>
> “MSVV” reference. Thanks for pointing out! This is the same algorithm described above in Mehta et al., but we realize that must have been confusing. Fixed.
>
> ** Addressing Technical Comments:
>
> -- “hyperparameter searching”:
> The networks we used in this work are fairly simple: dense layers with standard ReLu activation, and we use standard Adam optimizer. Simply choosing commonly recommended values for the parameters turn out to work well for the problems we looked at. In general, we agree with the reviewer’s point that hyperparameter searching can be important. For this particular work, our focus is to introduce the high-level ideas/framework and offer initial evidence that it can be effective, so we do not dwell much on the technical parts of ML in our discussion due to page limits. We also clarify that it is not the case that “we have no interests in extending ML techniques” in general. Indeed, we believe that for the future success of our approach on more open problems in online algorithms, it very much relies on the advances of ML in terms of neural network structure, optimization algorithms and training techniques. We also hope our work can motivate the design of new tools/techniques tailored for this direction.

---

### Official Review · AnonReviewer3 · 2019-10-23
**Official Blind Review #3**

**Rating:** 6

**Review:**

Update to the Review after the rebuttal from the Authors:
After carefully reviewing the responses by the authors especially on my concerns about the significance of solving an instance of a given problem and the improvement in the exposition of the ideas I would like to amend my earlier decision and recommend to accept. For completeness below is the original review.


This paper introduces a framework to learn to generate solutions to online combinatorial optimization problems with worst case guarantees. The framework as the authors claim eliminates the need for manual hard to solve instance/data creation, which is necessary to teach the model to provide the aforementioned worst case guarantees.  Therefore the main contribution of the paper can be said that this framework shows that it is possible to train a machine learning model, which can learn an algorithm to solve hard online combinatorial optimization problems and this training can be done without knowing much about the actual optimization problem domain. The only input required is the way to calculate the objective function of the actual problem. This contribution is demonstrated on two classes of problems: Ski-Rental and Fractional AdWords. The framework requires two neural networks one for solution generation agent and one for problem instance generation. These two networks are trained jointly from scratch and the underlying algorithm for the training is provided.

Although a generic framework that learns to solve online combinatorial optimization problems without domain knowledge is by itself a very motivational goal neither the paper successfully demonstrates that the framework the authors propose achieves this goal nor it explains well enough why one would take the machine learning approach to find good algorithms to such problems. Is it because the ML solution would be faster to compute with big instances? Is it because with the proposed approach one can curate sophisticated heuristic solutions when provable optimality is out of reach?

This paper should be rejected because proposed method demonstrates that an instance of one class of problems, Fractional Adwords, can be learned to solve without domain expertise, however fails to prove that the approach would be beneficial for any other instances of the same problem. Although they show that the Ski Rental problem can also be learned to solve though it is trivial and does not even use the framework the authors propose in its full extent, ie. problem instances are not generated by use of a machine learning model, which is one of main claims the authors are making. Therefore I do not find being able to solve this problem as a supporting evidence for the contributions claimed. In particular there is not any theoretical not experimental evidence that the approach would scale to any instances where a pure optimization approach would be slow to provide any meaningful solutions. I find this important because for combinatorial optimization usually scale matters a lot. While a small instance of a problem can be solve by a general purpose solver quickly a small increase in the problem size can turn out to be intractable. When proposing a machine learning approach to such problems I would expect the model to scale better than pure optimization approach so that there would be demonstrable benefit. Although the paper proposes an interesting framework I would argue that it is a “green apple” in the sense that authors need to motivate the approach better and expand the contribution beyond solving a particular instance. Authors acknowledge the fact that their experimental setup is rather limited in Appendix C.1, which I agree with and they also claim that there is a representation for a uniform algorithm for any number of advertisers for the AdWords problem, however they leave this as a future work, which I find unfortunate. I would recommend taking this direction rigorously and expand the contribution, which would prove to be a very sound contribution.

In order to clarify the exposition the following are some questions:
1. Authors call the approach YaoGAN due to its structural similarity to GANs. I understand the fact that they are training two neural networks in an alternating scheme, which is similar to the GAN training. How can one evaluate the solutions generated by this framework similar to how GAN generators are evaluated? Can one walk the latent distribution of the algorithm agent and draw insights, which might lead into tailoring some algorithms that would be appropriate for some input distribution although in general inferior in terms of worst case guarantees?

2. The main technical contribution claim needs to be elaborated.  I understand how the game theoretic framework is established but how does this manifest itself in the algorithm described in Section 3.1 needs more explanation.

3. Authors claim there are two shortcomings of the previous method proposed in Kong et. al 2018. They need to elaborate how their method overcomes these issues better.

4. Authors state that fractional relaxation of combinatorial mainly integer optimization problems, which is accurate. Yet their approach is only able to solve the fractional version of the AdWords problem. In addition I agree with the fact that although continuous relaxations to integer optimization problems might provide insightful directions they usually employed to to prove bounds on the heuristic approaches. Yet the authors stop at only solving this version with a machine learning approach, which does not hit the bar for me. I would have expected the authors to at least elaborate on why the current framework is not suitable for the non-relaxed problem. What are the shortcomings?

5.In Appendix A authors talk about no-regret dynamics, which are relevant. However, they state they loosely follow this approach. What does that entail? What kind of theoretical guarantees are given up due to not following this, a better exposition on this topic would help to support the claims.

6. In appendix C.2 authors provide additional plots for the Fractional AdWords problem. However, they retain from providing any intuition about them. In particular what is the conclusion to be drawn from Figure 5.  This needs more elaboration. Is this way of training results expected? What is the lesson learned?

7.In Figure 8 they provide example data from experience array. What are the significance of these examples? How they help us understand the problem instance generation was actually able to find interesting instances? What kind of dynamics are under covered? These are not directly revealed by only looking at the pictures one needs more explanation to support the claims.

**Experience Assessment:**

I have read many papers in this area.

**Review Assessment: Checking Correctness Of Derivations And Theory:**

I assessed the sensibility of the derivations and theory.

**Review Assessment: Checking Correctness Of Experiments:**

I assessed the sensibility of the experiments.

**Review Assessment: Thoroughness In Paper Reading:**

I read the paper thoroughly.

---

> ### Author Response · Authors · 2019-11-09
> **Overall response to reviewer #3**
>
>
> Thank you for your review. Please also see our high-level clarification above which we believe can help in better interpretation of our contribution. Some specific responses below:
>
> -- “proposed method demonstrates that an instance of one class of problems, Fractional Adwords, can be learned to solve without domain expertise, however fails to prove that the approach would be beneficial for any other instances of the same problem.”
>
> Please refer to our overall comments on this question (and also a few more details in reply to Reviewer#1’s similar question).
>
>
>
> -- Comment on scale / speed for large instances of combinatorial optimization:
>
>
> The point of this work is only to see if ML can find optimal algorithms, and not about doing it faster than the known theoretical algorithms. Note that this is not similar to the case of solving an offline combinatorial problem via integer programming or other solvers, since our problems are online, i.e., the instance is not known beforehand, so there is no comparison to such “general-purpose” solvers. Thus we don't compare to the running time of offline solvers, but to the worst-case competitive ratio of the optimal online algorithms. As mentioned in the comment, this approach may eventually lead to finding optimal or near-optimal algorithms for a problem (not an instance of a problem) for which no algorithm is known -- but this is outside the scope of this work future work.
>
> Again drawing the analogy of playing Go, the objective is mostly on training an agent that can make competitive moves rather than very fast moves, and there is no known “general-purpose” strategy to accomplish this.
>
> *Please also see reply to reviewer #2 on a similar question of evaluating against other methods*
>
>
> -- “Ski Rental problem can also be learned to solve though it is trivial and does not even use the framework the authors propose in its full extent, i.e. problem instances are not generated by use of a machine learning model, which is one of the main claims the authors are making.”
>
> Please see our high-level clarification on top.

---

> > ### Author Response · Authors · 2019-11-09
> > **Response to specific comments (1)--(7). part 2/2**
> >
> >
> > (4) We agree with the reviewer that in many cases there is a gap between solving the discrete problem and the fractional problem. In general it is an established approach to solve the fractional problem and use additional techniques such as rounding to fill the gap. As to AdWords, although the discrete problem naturally corresponds to the real world scenario, we do not consider fractional AdWords below the bar compared to discrete AdWords in terms of difficulty. The optimal CR bound and the adversarial distribution are the same for both cases, and the optimal algorithms basically have the same structure. One may arguably say that the optimal algorithm for the fractional problem has richer structure as in the fractional problem the action space is much larger as we can fractionally assign each ad to many advertisers.
> > As to the shortcomings of our techniques and why we pick the fractional problem, note that the GAN framework needs the computation of the discriminator network (i.e. the algorithm agent in our context) to be differentiable in order to update the generator network (i.e. the adversary in our context) during training. This poses difficulties if we ask the algorithm agent to make discrete decisions via sampling or rounding since it will not be differentiable. This doesn’t mean that our high-level framework (i.e. training the algorithm and adversary networks simultaneously) is doomed, since we can use other ML techniques (e.g. reinforcement learning) to implement our framework, but in general sampling and rounding will lead to much more work during training, so we pick the GAN structure in this work.
> >
> > (5) We know from theory that if the algorithm player runs a no-regret dynamic (e.g. MWU) and the adversary player responds with the worst input for the algorithm in each round, then the algorithm player converges to the optimal algorithm, and the uniform distribution over the adversary player’s responses gives the adversarial distribution. However, we cannot really follow this approach as the space of algorithms is infinite and we cannot run a MWU on this space, and in general it is also hard or impossible to find the absolute worst input in each round. In the practical framework, the algorithm player uses a neural network, and the adversary network tries its best to come up with a bad (but not necessarily worst) input each round. Thus we don’t have all the clean theoretical guarantees anymore, but the intuition should still largely hold (as our empirical result suggests).
> >
> >
> > (6) We updated the appendix to address this. See “Training convergence” in Appendix D.2
> > (7) We updated the appendix to address this. See “Adversarial distribution” in Appendix D.2

---

> > ### Author Response · Authors · 2019-11-09
> > **Response to specific comments (1)--(7). part 1/2**
> >
> >
> > Thanks for the comments, they are helpful in improving exposition.
> >
> > (1) In general, it can be difficult or impossible to quantitatively evaluate the solutions accurately (see the discussion of “Training convergence” in Appendix D.2). As to how we can make sense of the trained algorithm network and extract human-level insight or knowledge out of the neural network, interpretability in deep learning typically requires some domain expertise. In our work, we feed the algorithm network various inputs and inspect its outputs. We used AdWords in our work as demonstration because the optimal algorithm is known, so we can verify that our algorithm makes the right decision in different cases. An ultimate/ideal application of our approach is to facilitate the discovery of good algorithms for not so well-understood problems. That is, by inspecting the algorithm neural network’s behavior at carefully chosen inputs (e.g., the adversarially generated input instances during training), an expert can draw enough insight to extract the strategy out of the neural network into something humans can comprehend (i.e. “textbook” algorithm). Of course this step will require significant domain expertise, but the hope is that we can produce a good candidate algorithm (i.e. the neural network) without much domain expertise so the second phase is easier than drawing up a good algorithm from scratch. We note this is similar in the situation of GAN where domain experts draw insights on structure of latent space by inspecting the generator after training, or in the situation of chess, where top human players enhance their understanding of the game and come up with unconventional strategies via observing how the deep neural network plays.
> >
> >
> > (2) The game theoretical view says in the algorithm-adversary game, we want to find the min-max strategy of the algorithm player (i.e. optimal worst-case algorithm), and a known way to find such strategy is if the algorithm player runs a no-regret dynamic (e.g. multiplicative weights update), and the adversary player in each round plays the best response to the algorithm player’s strategy of that round. This suggests that to effectively train the algorithm agent, we should aim to find the best response (i.e. the input instance on which the algorithm performs the worst). This is fundamental in our framework since it tells what we want the adversarial network to search for (i.e. gives an objective to the Algorithm 1 in section 3.1). Retrospectively this seems obvious, but the previous approach (and also the predominant approach in classical ML) is to have a good training set upfront, i.e. a distribution over difficult input instances in our context. This is infeasible if we want a framework that can also work in the case where we don’t have such knowledge, and the no-regret dynamic approach allows us to get around this, and reduce the task to adaptively finding bad inputs through training. We accomplish this by using the adversary network and other techniques.
> >
> >
> > (3) The prior work of Kong et al. relies on a good training set of input instances a priori to effectively train the algorithm. The two shortcomings are [mentioned in the paragraph “Solving an Online Optimization Problem” at the end of Section 2]:  (1) it requires the knowledge of the adversarial input distribution (i.e. the maxmin player’s optimal strategy) and (2) even the adversarial distribution alone is not enough (e.g. the rock-paper-scissors example mentioned in the discussion) so additional human expertise is required to combine a high-entropy distribution to the adversarial distribution. Both shortcomings point to that it requires a significant amount of expert input for their approach to work, which is infeasible if the goal is to have a framework that can work for problems where we currently lack such knowledge. As discussed for the above question (2), we get around this issue by not trying to construct a good training set upfront, but adaptively come up with good training inputs as the training evolves. This is also in spirit the high-level strategy used in GANs and AlphaZero to get around the issue of no high-quality training datasets.
> > We have added a sentence at the end of the discussion “Solving an Online Optimization Problem” (end of Sec 2) to explain how we overcome the issue.

---

### Author Response · Authors · 2019-11-09
**High-level clarification 1/2**


We thank all the reviewers for the thorough review and feedback. We want to start with some high-level clarification.

Due to the nature of this work being at the intersection of ML and online algorithm design, there are various notions/results from the online algorithm side that are essential for ML audience to fully grasp the paper. Unfortunately, we had to be very brief (or skip completely) on most of them due to page limits. We expanded the appendix in our draft to make our result more self-contained to the ML audience. See Appendix A and D.

Although we are by no means claiming that our results have the same import as the breakthrough work on playing Go, it is very helpful to use the example in that domain as an analogy to what we are doing. For AlphaGo, its training required a large amount of high-quality examples of how top human players play the game. The framework of AlphaZero largely eliminates the requirement of such human-level expertise, and can learn competitive strategies from scratch via self-playing. The significance of AlphaZero over AlphaGo is largely in that the new framework can generalize to contexts/tasks where we don’t have high-quality examples for training. In our context, the previous work of Kong et al. demonstrates that ML agent can learn optimal algorithm if people train it with high-quality training instances, which requires significant human tuning. By contrast, our approach can work without such expertise, and thus opens the door for it to be useful to discover unknown algorithms.

We note that in the field of algorithm design, a problem being well-understood is not an indicator of its lack of difficulty or complexity, but also largely due to its importance and how much time researchers have devoted to it. In particular, AdWords is considered a very difficult problem in the field of online algorithms, and the MSVV [J. ACM’07] result that gave the algorithm and the optimality analysis is considered a breakthrough in the field. It is also an archetypical problem capturing the class of problems solvable by one of the most powerful techniques in this area -- primal-dual algorithms. More importantly, the fact that ski-rental and AdWords are well-understood doesn’t make the ML task any easier, since we don’t transfer any expert knowledge on these problems when we train networks. Thus, the fact that our framework can learn optimal algorithms on these problems suggests that it could also be effective on not so well-studied problems. Although the long-term goal is to draw insights from ML generated algorithms and design optimal algorithm for not well-studied problems, we look at AdWords and ski-rental in this work because our expert knowledge allows us to verify that our ML framework can be indeed effective in that the ML generated algorithms agree with our understanding of the optimal algorithms.

---

> ### Author Response · Authors · 2019-11-09
> **High-level clarification 2/2**
>
>
> Despite ski-rental’s simple structure, it is an important example showing why the previous approach of using a fixed training distribution upfront won’t work in general. In particular, if we fix any distribution over the season length, and use it as the training dataset, then the best algorithm against that distribution will be a deterministic strategy, which we know from theory cannot have a worst-case competitive ratio smaller than 2, whereas the optimal algorithm achieves ~1.58. This suggests we must switch to a framework where we adaptively come up with instances as we train the algorithm. Consider a more familiar example outside online algorithms, the game of rock-paper-scissors, if we want to train a player to learn the worst-case optimal strategy (i.e. playing the three moves uniformly at random), we cannot accomplish this by training against an adversary who plays a fixed (possibly randomized) strategy. No matter what the fixed strategy is (even the optimal uniform strategy), there is always a best deterministic strategy to counter (e.g. any strategy performs equally well against the fixed uniform strategy), so we shouldn’t expect our player to learn the optimal uniform strategy. We have to make the adversary adaptive during training.  Please see more comments on the significance of the ski rental problem in our response to Reviewer #1.
>
> While the Adwords problem is difficult (even for a small number of advertisers), and finding a worst-case algorithm for ski-rental is also not trivial to do with ML without any prior knowledge, we agree that our results can be much stronger if we also solve AdWords with arbitrary number of advertisers, and also solve more problems in online combinatorial optimization. We consider these tasks beyond the scope of this initial work, and will definitely pursue in future work. Training a network that can handle three advertisers may seem trivial compared to training a network that can simultaneously handle the cases of any number of advertisers, but it is not really the case. Again, using Go as an example, it is basically the difference between training a network that can excel at Go on the standard 19*19 board versus training a network that can excel (simultaneously) at Go on board of arbitrary size. Of course the latter is a much stronger result, but the fixed size task is by no means trivial. Similarly, AlphaZero only demonstrates its effectiveness on the examples of chess, shogi and Go rather than all possible tabletop games, (analogous to us only experimented with ski-rental and AdWords), but the results offer hope/indication that it can be effective in other settings.
>
>
> The main point of our work is to suggest a new framework in the context of ML+algorithm design, and is by nature exploratory. We admit there is a significant amount of work to be done to fulfill our long term goal of applying ML to facilitate the discovery of new algorithms that extend the frontier of  online combinatorial optimization. We think that this is a very meaningful direction to apply ML techniques in broader contexts, and can have significant practical impact. We hope both the ML and algorithm design communities can feel excited about this.

---

### Decision · Program_Chairs · 2019-12-19

**Decision:**

Reject

**Comment:**

The authors propose an intriguing way to designing competitive online algorithms. However, the state of the paper and the provided evidence of the success of the proposed methodology is too preliminary to merit acceptance.